# The Jag2/Notch1 signaling axis promotes sebaceous gland differentiation and controls progenitor proliferation

**Syeda Nayab Fatima Abidi[1]\*, Sara Chan[2], Kerstin Seidel[1], Daniel Lafkas[1], Louis Vermeulen[1], Frank Peale[2], Christian W Siebel[1]\***

[1]Department of Discovery Oncology, Genentech, San Francisco, United States; [2]Department of Research Pathology, Genentech, San Francisco, United States

## eLife assessment

This work aimed at deconstructing how sebaceous gland differentiation is controlled in adult skin. Using monoclonal antibodies designed to inhibit specific Notch ligands or receptors, the authors present **convincing** evidence that the Jag2/Notch1 signaling axis is a crucial regulator of sebocyte progenitor proliferation and sebocyte differentiation. The **valuable** findings presented here contribute to the growing evidence that Notch signaling is not only key during the development of the skin and its appendages but also regulates cell fate in adult homeostatic tissues. From a translational perspective, it is intriguing that the effect of Jag2 or Notch1 inhibition, which leads to the accumulation of proliferative stem/progenitor cells in the sebaceous gland and prevents sebocyte differentiation, is reversible.

**\*For correspondence:**
abidis1@gene.com (SNFA);
cwsiebel@gmail.com (CWS)

**Abstract** The sebaceous gland (SG) is a vital appendage of the epidermis, and its normal homeostasis and function is crucial for effective maintenance of the skin barrier. Notch signaling is a well-known regulator of epidermal differentiation, and has also been shown to be involved in postnatal maintenance of SGs. However, the precise role of Notch signaling in regulating SG differentiation in the adult homeostatic skin remains unclear. While there is evidence to suggest that Notch1 is the primary Notch receptor involved in regulating the differentiation process, the ligand remains unknown. Using monoclonal therapeutic antibodies designed to specifically inhibit of each of the Notch ligands or receptors, we have identified the Jag2/Notch1 signaling axis as the primary regulator of sebocyte differentiation in mouse homeostatic skin. Mature sebocytes are lost upon specific inhibition of the Jag2 ligand or Notch1 receptor, resulting in the accumulation of proliferative stem/progenitor cells in the SG. Strikingly, this phenotype is reversible, as these stem/progenitor cells re-enter differentiation when the inhibition of Notch activity is lifted. Thus, Notch activity promotes correct sebocyte differentiation, and is required to restrict progenitor proliferation.

## Introduction

The skin is a vital organ that acts as a protective barrier against the external environment, and safeguards against fluid loss. An important component of this barrier function is the presence of a complex mixture of oils, known as sebum, which is produced by the SGs. SGs are part of the epidermis and are typically associated with the hair follicle. These acinar structures have two cell types: basal stem or progenitor cells which encase the differentiated sebocytes (*Figure 1a*). Sebocyte differentiation begins at the proximal tip of the of the SG, with maturing sebocytes moving upwards, enlarging, accumulating lipids, and ultimately undergoing a highly regulated and specialized form of cell death in

which they release their lipid contents into the sebaceous duct (*Figure 1b*; *Kretzschmar et al., 2014*; *Schneider and Paus, 2010*). This process requires the constant turnover of sebocytes, which occurs over a period of 7–14 d in mice (*Jung et al., 2015*). Both over- and underproduction of sebum have been linked to various skin disorders including acne or dry skin (*Al-Zaid et al., 2011*; *Binczek et al., 2007*; *Karnik et al., 2009*; *Lovászi et al., 2017*; *Rittié et al., 2016*; *Seiffert et al., 2007*; *Shi et al., 2015*; *Smith and Thiboutot, 2008*; *Stenn et al., 1999*), and rare sebaceous carcinomas constitute aggressive tumors leading to high mortality (*Buitrago and Joseph, 2008*; *Nelson et al., 1995*), thus SG number and function have to be tightly regulated for proper skin function.

The Notch signaling pathway is one of the most studied regulators of cell fate decisions, and is known to be widely involved in epidermal differentiation. The pathway consists of multiple ligands and receptors that typically form a signaling axis in pairs. Notch signaling can regulate cell fate by either inducing or inhibiting differentiation, or by making binary cell fate decisions (*Wilson and Radtke, 2006*). Classically, these functions of Notch signaling have been studied during development, but increasing evidence suggests that the Notch pathway is also involved in regulating cell fate and cell states in adult homeostatic tissues (*Ables et al., 2011*; *Lafkas et al., 2015*; *Mosteiro et al., 2023*; *Sato et al., 2012*; *Siebel and Lendahl, 2017*).

While it is known that Notch signaling is not required for embryonic development of the epidermis, it is essential for the postnatal maintenance of the hair follicles and the SGs (*Watt et al., 2008*). However, the precise role of Notch signaling in adult sebocyte differentiation has not been comprehensively investigated, with most studies examining irreversible deletions of the Notch pathway components in the embryonic ectodermal lineages. While these studies report SG defects, it remains unclear whether these defects are due to a direct effect on the SGs, or whether they are a consequence of general skin defects also observed in these models. For example, SGs are absent in mice with embryonic pan-Notch deletions such as *Rbpj* (*Blanpain et al., 2006*), gamma-secretase, *Notch1Notch2*, and *Notch1Notch2Notch3*, and are severely reduced in *Notch1* and *Notch1Notch3* embryonically-deleted skin (*Pan et al., 2004*), while deletion of *Notch2* alone does not affect the SG (*Pan et al., 2004*). Consistent with the constitutive deletions, loss of *Rbpj* in adult SGs also results in missing sebocytes, while loss of *Notch1* in the adult SGs results in miniaturized lobes that still contain some differentiated sebocytes (*Veniaminova et al., 2019*). Interestingly, activation of *Notch1* in the adult skin results in enlarged SGs (*Estrach et al., 2006*). Conversely, for the Notch pathway ligands, embryonic and adult deletion of *Jag1* (*Estrach et al., 2006*), and embryonic deletion of *Dll1* (*Estrach et al., 2008*) results in normal SG morphology. Collectively, these data suggest that Notch1 is the dominant Notch receptor involved in regulating sebocyte differentiation, however, it remains unclear which ligand is required.

In our previous work, we observed that systemic inhibition of Jag2 using monoclonal therapeutic antibodies resulted in SG defects in adult mice (*Lafkas et al., 2015*). Given that our therapeutic antibodies selectively, potently, and transiently inhibit each of the distinct Notch receptors and ligands (*Tran et al., 2013*; *Wu et al., 2010*; *Yu et al., 2020*), they constitute ideal tools to dissect the contribution of each of these pathway members to sebocyte differentiation in adult homeostatic skin. Leveraging the use of these antibodies, we demonstrate that specific inhibition of the Jag2 ligand or Notch1 receptor both result in the loss of mature sebocytes in the SG, establishing the Jag2/Notch1 signaling axis as a crucial regulator of sebocyte differentiation in adult homeostatic skin. The loss of mature sebocytes in the SG is concomitant with an accumulation of cells with a basal phenotype, forming epithelial remnants in the SG. Cells in these epithelial remnants are actively proliferating and express stem/progenitor markers indicating that sebocyte differentiation is halted, while stem/progenitor numbers are increased. Importantly, this phenotype is reversible, as these epithelial cells re-enter differentiation with the return of Notch activity. Thus, Notch activity is required in sebocyte stem/progenitor cells for their proper differentiation, and its inhibition locks these cells in a reversible progenitor state.

## Results

### Jag2 is the dominant Notch signaling ligand involved in regulating sebocyte differentiation

To investigate the role of Notch signaling in the homeostatic skin, we treated 8- wk-old mice with a single dose of the various antagonizing antibodies (anti-Notch1, anti-Notch2, anti-Jag1, anti-Jag2,

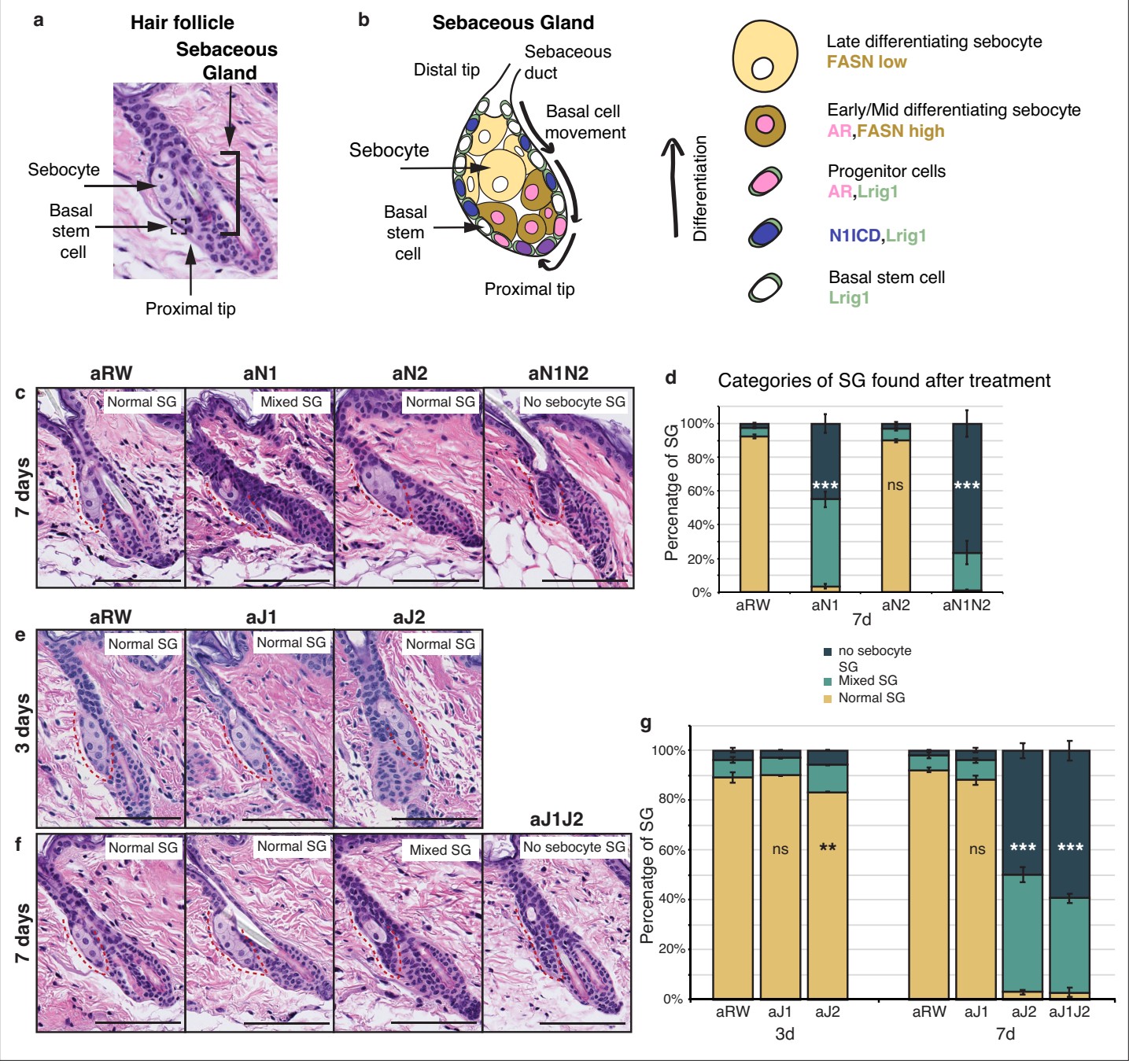

**Figure 1.** Jag2 is the dominant ligand involved in regulating sebocyte differentiation. (**a**) A representative image of the homeostatic hair follicle, including its associated sebaceous gland (SG). (**b**) A detailed schematic of the SG showing the outer basal stem cell layer encasing the differentiated sebocytes. The schematic also shows the gene expression in different regions of the SG. (**c**) Representative hematoxylin and eosin (H&E) images of SGs from mice (n=5 each) treated with aRW, aN1, aN2, and aN1N2, 7 d post-treatment. (**d**) Quantification of type of SG found after each treatment. The SGs were divided into three categories: normal SGs containing a characteristic number of sebocytes (normal SG), SGs containing a mix of sebocytes and basal-like cells (mixed SG), and SGs containing no sebocytes and only basal-like cells (no sebocyte SG). For (**c**), (**e**), and (**f**), each SG panel is labeled with where the SG is normal, mixed, or no sebocyte SG. p-values: aN1=2.02E-123, aN2=0.426, aN1N2=3.85E-114. Total n of SGs quantified per treatment: aRW = 425, aN1=298, aN2=343, aN1N2=217. (**e**) Representative H&E images of SGs from mice (n=5 each) treated with aRW, aJ1, and aJ2, 3 d post-treatment. (**f**) Representative H&E images of SGs from mice (n=5 each) treated with aRW, aJ1, aJ2 and aJ1J2, 7 d post-treatment. (**g**) Quantification of type of SG found after each treatment. 3 d: p-values: aJ1=0.409, aJ2=0.004. 7 d: p-values: aJ1=0.057, aJ2=7.61E-151, aJ1J2=6.23E-169. Total n of SGs quantified per treatment: 3 d, aRW = 851, aJ1=944, aJ2=760, 7 d, aRW = 491, aJ1=499, aJ2=388, aJ1J2=461. Chi-square test used for statistical analysis. All treatments were compared against aRW. Error bars represent SEM. Scale bars are 100 μm.

*Figure 1 continued on next page*

*Figure 1 continued*

The online version of this article includes the following source data and figure supplement(s) for figure 1:

**Source data 1.** Source data for *Figure 1*.

**Figure supplement 1.** Sebaceous ducts remain unaffected after Notch inhibition.

**Figure supplement 1—source data 1.** Source data for *Figure 1—figure supplement 1*.

and the isotype control antibody anti-Ragweed) alone or in combination. We then examined the dorsal, resting phase (telogen) skin at 3, 7, and 14 d post-treatment. We first confirmed that our antibodies could reproduce the requirement of Notch1 in regulating sebocyte differentiation. To this end, we treated mice with Notch1 (aN1) and Notch2 (aN2) blocking antibodies and examined the SGs at 7 d post-treatment. Loss of mature sebocytes was observed specifically after aN1 treatment, while the aN2-treated SGs showed normal morphology (*Figure 1c and d*). However, the combined treatment of aN1N2 had a more pronounced effect on SG morphology as compared to aN1 alone (*Figure 1c and d*), indicating that Notch2 also contributes to regulating sebocyte differentiation, possibly as a compensatory mechanism after inhibition of the dominant Notch1 receptor.

Next, we investigated which ligand formed the signaling pair with Notch1. Loss of mature sebocytes was observed specifically after treatment with the Jag2 blocking antibody (aJ2), but not after treatment with the Jag1 blocking antibody (aJ1) at 7 d post-treatment (*Figure 1e and f*). For aJ2 treatment, the loss of sebocytes began at 3 d post treatment, but affected only a small proportion of the SGs at this time point (*Figure 1g*). However, by 7 ds post-treatment, most SGs had either completely lost all sebocytes, or had only some sebocytes remaining (*Figure 1g*). The loss-of-sebocyte phenotype was most pronounced with a combined treatment of Jag1/Jag2 blocking antibodies (aJ1J2) (*Figure 1g*), suggesting that while Jag2 is the primary Notch ligand involved in regulating sebocyte differentiation, Jag1 also plays a minor role in this process, but is unable to produce a phenotype on its own. In conclusion, these data indicate that the Jag2-Notch1 signaling axis is the dominant Notch ligand-receptor pair required for sebocyte differentiation in the adult skin.

We also examined the percentage of SGs consisting of bursting sebocytes releasing sebum as a proxy of a functional sebaceous duct. There were no significant differences between treatments (*Figure 1—figure supplement 1a and b*), hinting at a functionally intact sebaceous duct.

## Notch is active in the sebaceous gland stem cells

We next characterized the expression of the relevant receptor and ligand in the SG to develop a spatial map of Notch activity in this tissue. Both the Notch receptors and their ligands are transmembrane proteins that interact with each other in neighboring cells to activate the pathway. To examine this interaction, we performed a triple stain for the cleaved (active) form of the Notch1 intracellular domain (ICD), and Notch1 and Jag2 in situ hybridization (ISH) probes at 3 d post antibody treatment. In the control treated mice, Notch1 ICD was observed in the basal stem cell compartment of the SG, but not in the differentiating sebocytes (*Figure 2a–c*, cell types based on morphology). We saw a similar pattern of N1ICD-positive cells for aJ1 treatment (*Figure 2d–f*). Notch1 signaling was not active in all basal cells, as only ~50% of them were positive for N1ICD after control and aJ1 treatment (*Figure 2g*). Consistent with previous studies, the majority of these N1ICD + cells were present near the proximal tip of the SG (*Figure 2—figure supplement 1a–c*; *Veniaminova et al., 2019*), where the initial sebocyte differentiation has been proposed to occur (*Kretzschmar et al., 2014*). However, after aJ2 treatment, Notch1 activity (ICD staining) was absent, or observed only at very low levels in the basal stem cells, while they still expressed Notch1 and Jag2 mRNA (*Figure 2h–j*). Strikingly, we noticed the expression of both Notch1 ISH and Jag2 ISH in the same cell on most N1ICD + cells (*Figure 2k*). Interestingly, a majority of all basal cells (including N1ICD- cells), expressed both N1 and Jag2 mRNA (*Figure 2—figure supplement 1d*). Our data shows that the vast majority of SG basal stem cells express both the ligand and receptor, but only some basal stem cells experience active Notch signaling at any one point in time, as evidenced by the presence of N1ICD. This could be a technical limitation as the triple staining only captures a temporal snapshot of the basal stem cells, with the N1 + J2 mRNA positive cells going on to express N1ICD later, or it could hint towards a more complex regulatory mechanism involved in activating Notch signaling.

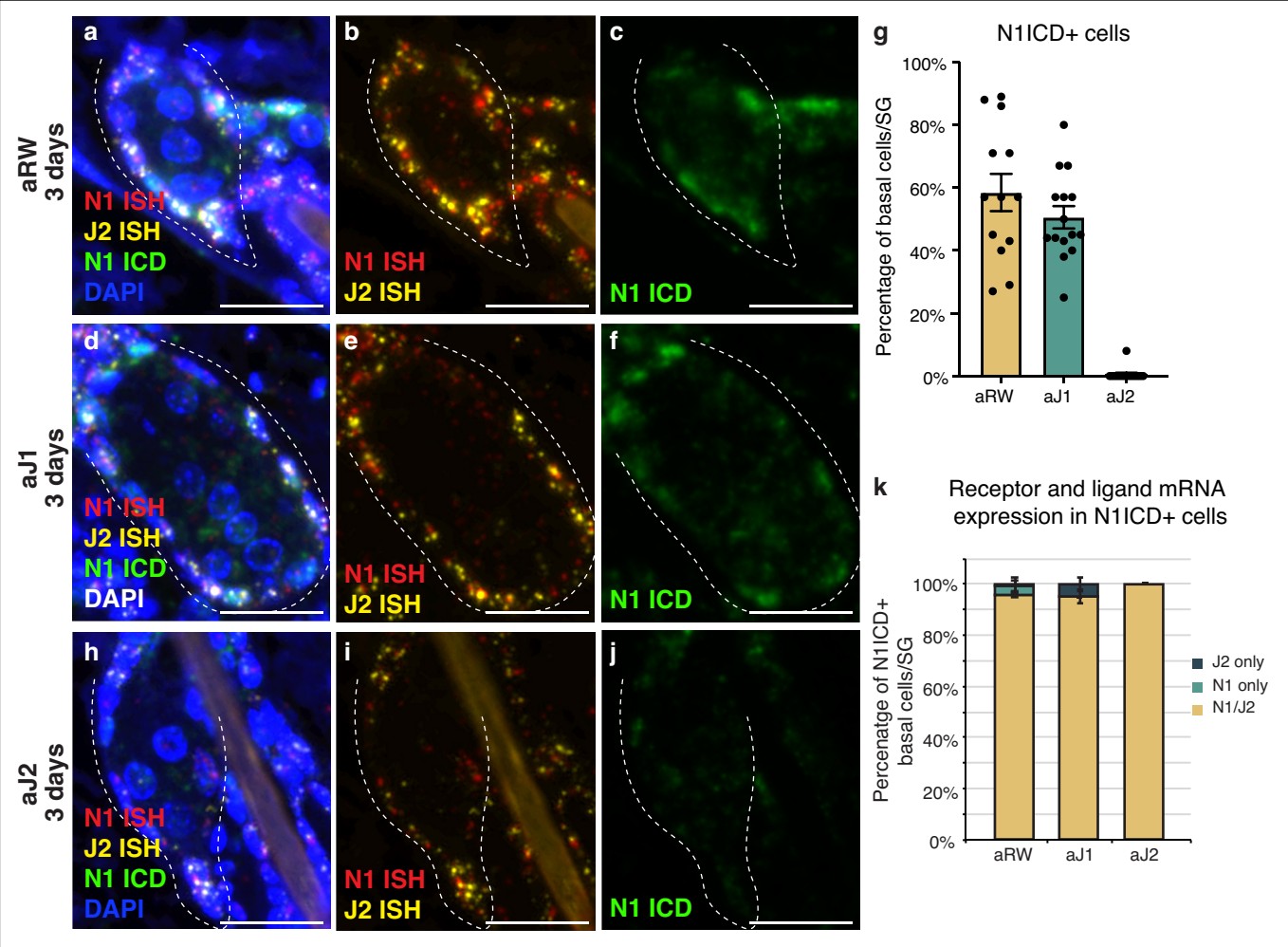

**Figure 2.** Notch is active in the sebaceous gland stem cells. (**a–c**) Representative triple stain images for N1ICD, Notch1, and Jag2 mRNA in sebaceous glands (SGs) from mice (n=5 each) treated with aRW, 3 d post-treatment. (**a**) Four channels showing N1ICD, N1 ISH, J2 ISH, and DAPI. (**b**) Two channels showing N1 ISH and J2 ISH. (**c**) One channel showing N1ICD. (**d-f**) Representative triple stain images for N1ICD, Notch1, and Jag2 mRNA in SGs from mice (n=5 each) treated with aJ1, 3 d post-treatment. (**d**) Four channels showing N1ICD, N1 ISH, J2 ISH, and DAPI. (**e**) Two channels showing N1 ISH and J2 ISH. (**f**) One channel showing N1ICD. (**g**) Quantification of the percentage of N1ICD positive (N1ICD+) basal stem cells in SGs from mice (n=5 each) treated with aRW, aJ1, and aJ2, 3 d post-treatment. Percentage was calculated by dividing the number of N1ICD+ basal stem cells by the total number of basal stem cells in each SG. (**h–j**) Representative triple stain images for N1ICD, Notch1, and Jag2 mRNA in SGs from mice (n=5 each) treated with aJ2, 3 d post-treatment. (**h**) Four channels showing N1ICD, N1 ISH, J2 ISH, and DAPI. (**i**) Two channels showing N1 ISH and J2 ISH. (**j**) One channel showing N1ICD. (**k**) Quantification of what percentage of the N1ICD+ basal stem cells express both N1 ISH and J2 ISH, only N1 ISH, or only J2 ISH. (**g and k**) Total n of SGs quantified per treatment: aRW=13, aJ1=15, aJ2=15. Error bars represent SEM. Scale bars are 25 µm.

The online version of this article includes the following source data and figure supplement(s) for figure 2:

**Source data 1.** Source data for *Figure 2*.

**Figure supplement 1.** Notch is active in the sebaceous gland stem cells.

**Figure supplement 1—source data 1.** Source data for *Figure 2—figure supplement 1*.

Notch activity, as well as Notch1 and Jag2 mRNA, were also observed in the interfollicular epidermis (IFE) and other cells of the hair follicle (*Figure 2—figure supplement 1e and f*). While there is a reduction in Notch-active cells in these other regions, it does not appear to significantly impact the rest of the skin. There were no significant differences in the width of the IFE or the adipocyte layer between treatments (*Figure 2—figure supplement 1g and h*). This together with the histological appearance (*Figure 2—figure supplement 1i and j*) of these regions suggests that proliferation and differentiation in these compartments remain unaffected.

## Loss of Notch activity in the SG stem cells inhibits sebocyte differentiation

To further detail the effect of Notch inhibition on sebocyte differentiation, we examined the expression of mature sebocyte markers. Adipophilin (Adipo) is expressed in all mature sebocytes (*Frances and Niemann, 2012*; *Ostler et al., 2010*) and Fatty acid synthase (FASN) is expressed in mid- and late-differentiating sebocytes, with FASN levels decreasing in the most mature sebocytes (*Cottle et al., 2013*; *Figure 1b*). Since N1ICD staining disappears at 3 d post antibody treatment, we examined the mature sebocyte markers at this time point. While N1ICD staining was specifically lost in the SG basal stem cells of aJ2-treated skin, mature sebocytes expressing FASN were still observed at this timepoint (*Figure 2j*, *Figure 3a and b* and *Figure 3—figure supplement 1a*). All cells that express FASN also express adipophilin, but since FASN levels decrease with sebocyte maturity, we also examined and focused on adipophilin to mark all sebocytes (*Figure 3c and d* and *Figure 3—figure supplement 1b–e*). The number of cells that expressed adipophilin was not significantly different between treatments at 3 d post antibody treatment (*Figure 3h*). At 7 d post antibody treatment, control and aJ1 treated skin showed normal SG morphology and sebocyte marker expression (*Figure 3e* and *Figure 3—figure supplement 1f, h and i*), but both aJ2 and aJ1J2 SGs had lost sebocyte marker expression, and the SG was filled with cells with a basal phenotype (basal-like cells) (*Figure 3f* and *Figure 3—figure supplement 1g, j and k*). Consistently, the number of cells expressing adipophilin were significantly lower for aJ2 and aJ1J2 treatment (*Figure 3i*). Interestingly, we noticed that some of these affected SGs still contained a few mature sebocytes (*Figure 3f* and *Figure 3—figure supplement 1g*), which were found at the distal end near the sebaceous duct. The location of these remaining sebocytes suggests that existing mature sebocytes are not affected by the Notch blockade, and go through their normal differentiation process, eventually bursting and releasing the sebum (*Figure 3g*). Thus, we propose that Notch blockade inhibits differentiation at the basal stem cell level or in a sebocyte progenitor.

## Notch activity in the SG stem cells is required to prevent unregulated progenitor proliferation

To determine whether the epithelial cells that filled the affected SGs at 7 d post-treatment were stem/progenitor cells, we examined them for stem and early differentiation markers. Lrig1-positive cells form a distinct stem cell compartment that maintains the SG and the upper part of the hair follicle (*Frances and Niemann, 2012*; *Niemann and Horsley, 2012*; *Page et al., 2013*), allowing us to use Lrig1 as a marker for the basal stem cells of the SG, while Androgen Receptor (AR) can be used as an early marker of sebocyte differentiation (*Cottle et al., 2013*; *Figure 4a and b* and *Figure 4—figure supplement 1a and b*). We noticed that the AR-expressing cell population could be divided into two groups: a basal stem cell population that co-expressed Lrig1, but did not express FASN (arrowheads in *Figure 4a and c*), and an early differentiating sebocyte population that expressed FASN, but did not express Lrig1 (arrows in *Figure 4a and c*). We hypothesize that the Lrig1+/AR + population is a progenitor cell population, in addition to the Lrig1 + stem cells, similar to the recently identified transitional basal cell population in the SG (*Veniaminova et al., 2023*). There were no significant differences in the number of Lrig1 positive stem cells per SG, or the AR-expressing progenitor population between treatments at 3 d post-treatment (*Figure 4d and e*). By 7 d post-treatment, however, the basal-like cells that filled the SG were all positive for Lrig1 (*Figure 4f and g* and *Figure 4—figure supplement 1e*), with the total number of Lrig1 positive cells per SG increasing significantly for aJ2 and aJ1J2 treated SGs (*Figure 4h*), while SGs after aJ1 treatment showed normal morphology and marker expression (*Figure 4—figure supplement 1c and d*). The proportion of AR-expressing cells was not significantly different between treatments (*Figure 4i*). These results indicate that the basal-like cells that fill the SG after blocking Notch signaling by aJ2 and aJ1J2 treatment, are stem/progenitor cells. Additionally, we examined the SGs for their proliferative capacity to confirm stem/progenitor function. In a normal SG, proliferation is restricted to the basal stem cells (*Figure 4j*). As expected, proliferation was also restricted to the basal stem cells in aJ1-treated SGs (*Figure 4—figure supplement 1f and g*). Remarkably, most of the basal-like cells in the aJ2-treated SGs were proliferative, while the rare mature sebocytes remaining in the SG were non-cycling (*Figure 4k*). As the SGs are mostly filled with basal-like cells at this time (*Figure 4h*), the total number of proliferating cells per SG was significantly higher (*Figure 4l*).

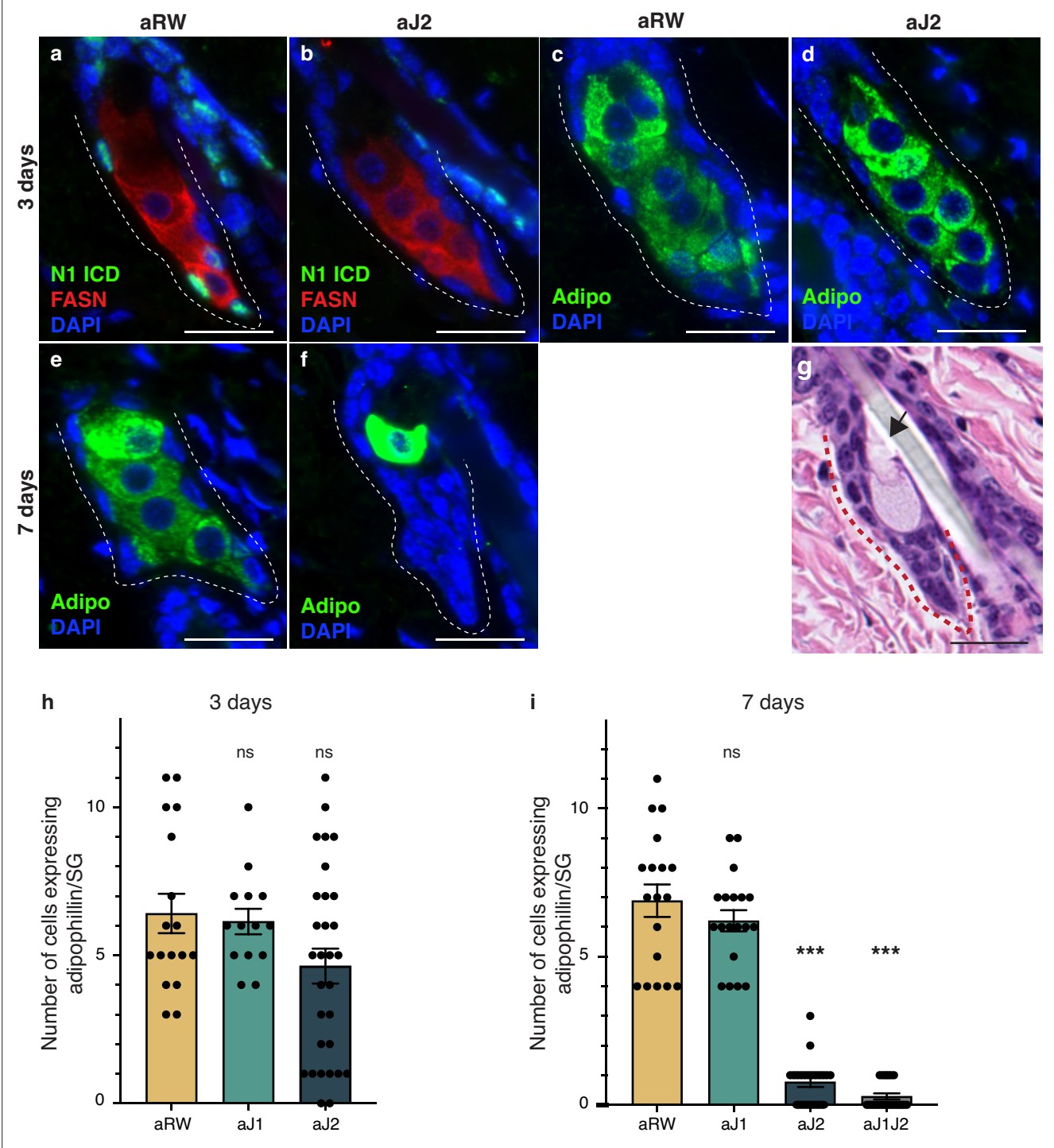

**Figure 3.** Loss of Notch activity in the sebaceous gland (SG) stem cells inhibits sebocyte differentiation. (**a, b**) Representative co-stain images for N1ICD and fatty acid synthase (FASN) in SGs from mice (n=5 each) treated with aRW (**a**) and aJ2 (**b**), 3 d post-treatment. (**c, d**) Representative adipophiin staining in SGs from mice (n=5 each) treated with aRW (**c**) and aJ2 (**d**), 3 d post-treatment. (**e, f**) Representative adipophilin staining in SGs from mice (n=5 each) treated with aRW (**e**) and aJ2 (**f**), 7 d post-treatment. (**g**) A representative hematoxylin and eosin (H&E) image of an SG from mice (n=5 each) treated with aJ2, 7 d post-treatment. Arrow points to a bursting sebocyte releasing sebum into the sebaceous duct. (**h**) Quantification of the number of cells expressing adipophilin in each SG, 3 d post-treatment with aRW, aJ1, and aJ2. p-values: aJ1=0.749, aJ2=0.062. Total n of SGs quantified per treatment: aRW = 17, aJ1=14, aJ2=30. (**i**) Quantification of the number of cells expressing adipophilin in each SG, 7 d post-treatment with aRW, aJ1, aJ2, and aJ1J2. p-values: aJ1=0.301, aJ2=4.13E-14, aJ1J2=4.06E-14. Total n of SGs quantified per treatment: aRW = 18, aJ1=19, aJ2=22, aJ1J2=21. Student's t-test used for statistical analysis. All treatments were compared against aRW. Error bars represent SEM. Scale bars are 25 μm.

*Figure 3 continued on next page*

*Figure 3 continued*

The online version of this article includes the following source data and figure supplement(s) for figure 3:

**Source data 1.** Source data for *Figure 3*.

**Figure supplement 1.** Loss of Notch activity in the sebaceous gland (SG) stem cells inhibits sebocyte differentiation.

Overall, these data suggest that inhibition of Notch activity by aJ2 treatment retains the stem and progenitor (Lrig1 + and Lrig1+/AR+, respectively) cells in their immature proliferative state and prevents differentiation.

## The block in sebocyte differentiation is lifted upon recovery of Notch activity

The therapeutic antibodies employed do not inhibit Notch signaling permanently, as the antibodies eventually become cleared from the animal's system. To determine whether the loss of sebocyte phenotype was reversible, we examined the SGs 14 d post single-dose treatment. Intriguingly, mature sebocytes begin to recover at this time point (*Figure 5a and b*, compared with *Figure 1f and g*). We hypothesized that the sebocyte recovery must be due to the return of Notch activity after antibody washout. To test this, we examined the SGs for Notch activity at 7 d post-treatment, since the return of Notch activity must precede the recovery of mature sebocytes. Fittingly, we observed N1ICD expression return in some of the basal-like cells at this time (*Figure 5c and d* and *Figure 5—figure supplement 1a–d*). The percent of N1ICD positive cells per SG strongly increased from 1% at 3 d post-treatment to 29% at 7 d post-treatment (*Figure 5e*), even though the percentage of N1ICD positive cells per SG remained significantly lower than control for aJ2 and aJ1J2 treated SGs at this time (*Figure 5—figure supplement 1d*). Next, we examined the SGs at 14 d post-treatment for mature sebocyte markers to confirm the sebocyte recovery. Indeed, we saw the return of cells expressing adipophilin in the aJ2 and aJ1J2 treated SGs (*Figure 5f and g* and *Figure 5—figure supplement 1e–h*). There was an overall increase in the number of these cells from day 7 to day 14 (*Figure 5h*), even though the number of these cells remained significantly lower in these SGs compared to controls (*Figure 5—figure supplement 1h*). Interestingly, the majority of the adipophilin-expressing cells were found in the proximal third of the SG (51% for aJ2 treatment and 66% for aJ1J2 treatment), consistent with the initiation of sebocyte differentiation at the proximal tip. However, there was a significant proportion of these cells found in the middle third (22% and 25%, respectively), and distal third (27% and 9%, respectively) of the SG. This could be due to the newly differentiated cells moving through the SG in a proximal to distal direction, as is the case during normal homeostasis. Alternatively, the stem cells could also be differentiating at sites other than just the proximal tip, as previously demonstrated by multi-color lineage tracing (*Andersen et al., 2019*). We further examined the SGs at 14 d post-treatment for their AR expression, and confirmed that it had also been restored to its normal pattern (*Figure 5i and j* and *Figure 5—figure supplement 1i–l*). We also found that the average number of AR-expressing cells decreased from day 7 to day 14 post aJ2 and aJ1J2 treatment (10–2.95, and 8.56–5.55, respectively) (*Figure 5k*), returning to a more homeostatic state. Together, these data indicate that Notch inhibition does not result in a permanent cell fate switch, but maintains the stem/progenitor state, allowing the recovery of the differentiation process with the restoration of Notch activity.

## Discussion

Based on our findings, we propose that the Jag2/Notch1 signaling axis is essential for correct sebocyte differentiation in homeostatic dorsal skin, and that inhibition of this signaling retains the basal stem and progenitor cells in a proliferative state, and blocks further differentiation. Thus, Notch signaling is required to prevent unregulated stem/progenitor proliferation, and induction of the sebocyte differentiation program. Moreover, Notch inhibition resulting in complete loss of mature sebocyte differentiation is a reversible phenotype, indicating that there remains a functional progenitor pool present during the studied timeframe.

Here, we have leveraged the use of monoclonal therapeutic antibodies designed to inhibit each of the distinct Notch receptors or ligands to study the role of Notch signaling in adult homeostatic tissue.

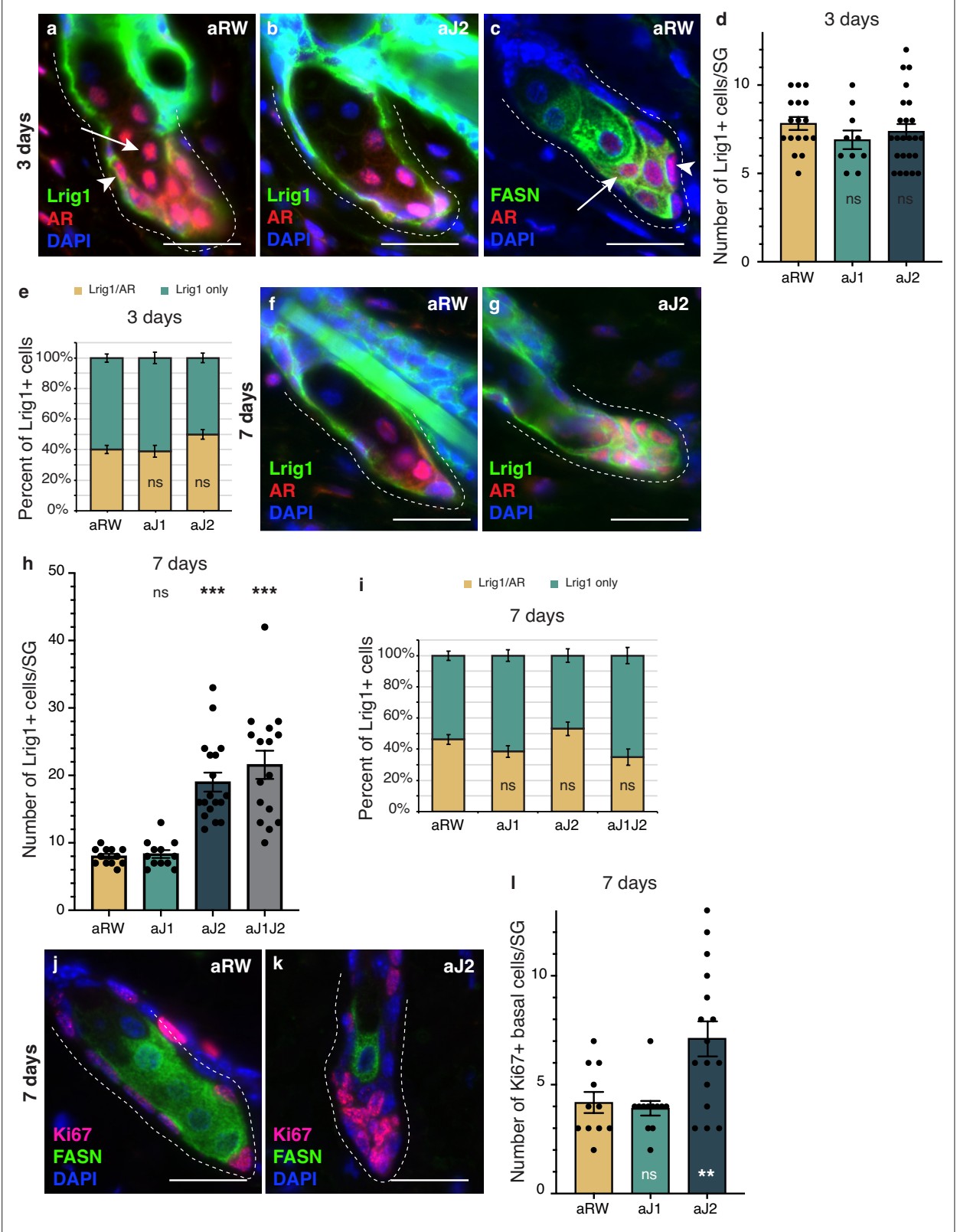

**Figure 4.** Notch activity in the sebaceous gland (SG) stem cells is required to prevent unregulated progenitor proliferation. (**a,b**) Representative co-stain images for Lrig1 and androgen receptor (AR) in SGs from mice (n=5 each) treated with aRW (**a**) and aJ2 (**b**), 3 d post-treatment. (**c**) A representative co-stain image for fatty acid synthase (FASN) and AR in an SG from mice (n=5 each) treated with aRW, 3 d post-treatment. Arrows point to sebocytes, and arrowheads point to progenitor cells. (**d**) Quantification of the number of cells expressing Lrig1 in each SG, 3 d post-treatment with aRW, aJ1. and

*Figure 4 continued on next page*

*Figure 4 continued*

aJ2. p-values: aJ1=0.152, aJ2=0.450. (**e**) Quantification of what percentage of the Lrig1 + cells express AR, 3 d post-treatment with aRW, aJ1, and aJ2. p-values: aJ1=0.789, aJ2=0.028. (**d and e**) Total n of SGs quantified per treatment: aRW = 17, aJ1=10, aJ2=24. (**f,g**) Representative co-stain images for Lrig1 and AR in SGs from mice (n=5 each) treated with aRW (**f**) and aJ2 (**g**), 7 d post-treatment. (**h**) Quantification of the number of cells expressing Lrig1 in each SG, 7 d after treatment with aRW, aJ1, aJ2, and aJ1J2. p-values: aJ1=0.628, aJ2=9.06E-07, aJ1J2=7.88E-06. (**i**) Quantification of what percentage of the Lrig1 + cells express AR, 7 d after treatment with aRW, aJ1, aJ2, and aJ1J2. p-values: aJ1=0.117, aJ2=0.248, aJ1J2=0.091. (**h and i**) Total n of SGs quantified per treatment: aRW = 12, aJ1=12, aJ2=18, aJ1J2=16. (**j,k**) Representative co-stain images for Ki67 and FASN in SGs from mice (n=5 each) treated with aRW (**j**) and aJ2 (**k**), 7 d post-treatment. (**l**) Quantification of the number of cells expressing Ki67 in each SG, 7 d after treatment with aRW, aJ1, and aJ2. p-values: aJ1=0.652, aJ2=0.009. Total n of SGs quantified per treatment: aRW = 11, aJ1=12, aJ2=16. Student's t-test used for statistical analysis. All treatments were compared against aRW. Error bars represent SEM. Scale bars are 25 μm.

The online version of this article includes the following source data and figure supplement(s) for figure 4:

**Source data 1.** Source data for *Figure 4*.

**Figure supplement 1.** Notch activity in the sebaceous gland (SG) stem cells is required to prevent unregulated progenitor proliferation.

We have shown that Jag2 is the hitherto unknown ligand involved in regulating sebocyte differentiation. Inhibition of Notch signaling using Jag2 blocking antibodies results in the loss of mature sebocytes, with the resulting SG being filled with basal-like cells forming 'finger-like' epithelial remnants. A similar phenotype has been described by a previous study (*Veniaminova et al., 2019*) that used Lrig1-CreERT2 to irreversibly knock out RBPJ in adult homeostatic skin. This strategy enabled inhibition of Notch signaling specifically in the Lrig1 + stem cell population that normally maintains the SGs, but preserved it in the rest of the hair follicle. Interestingly, the authors saw two contradictory phenotypes as a result: overall loss of SG lobes, replaced by the finger-like epithelial remnants, as well as the presence of persistent SGs that were enlarged. The authors showed that over time, patches of Rbpj mutant cells extended out of their niche into the IFE, where Lrig1 is not expressed, and that loss of RBPJ in the IFE led to enlarged SGs. They argue that while Notch signaling promotes sebocyte differentiation in the SG stem cells, it indirectly suppresses these glands from the IFE. Our results did not show the enlarged SG phenotype, and we only observed the finger-like epithelial remnants. While injections of the Notch blocking antibodies are systemic, we only observed a reduction in the number of Notch-active cells in the IFE, but not a complete loss. This could explain why we didn't observe the enlarged SG phenotype. Additionally, we observed the phenotype as early as 3 d post antibody treatment, while Veniaminova et al., observed the SGs 10 wk after tamoxifen injection. These differences are likely due to the different methodologies used in the two studies.

Several studies indicate that Notch signaling is essential for the postnatal maintenance of SGs (*Blanpain et al., 2006*; *Estrach et al., 2008*; *Estrach et al., 2006*; *Pan et al., 2004*; *Watt et al., 2008*), but so far only Veniaminova et al., have specifically examined SGs in the adult homeostatic skin. They observed that the stem/progenitor and differentiation markers were intermingled in the finger-like remnants forming an unnatural hybrid state, neither staying in a true progenitor state nor differentiating. In contrast, we observed a complete block in differentiation, with the stem/progenitor cells being locked in an immature proliferative state. These observations suggest that antagonistic antibodies may be able to achieve a more complete inhibition of Notch signaling in the SG stem/progenitor compartment.

Importantly, we were able to show that the loss-of-sebocyte phenotype in the SG is reversible. Having used therapeutic antibodies to block Notch signaling, the inhibition of the signaling pathway was not permanent. As Notch activity returned to the SG after antibody washout, sebocyte differentiation also recovered. These data indicate that the loss of Notch signaling does not impact the stem/progenitor potential of the SG, but prevents further differentiation. Indeed, a functional progenitor pool accumulates in the SG, primed for differentiation, as soon as Notch signaling becomes active. A recent study by Veniaminova et al., has shown that Lrig1-CreERT2 used to irreversibly knock out *Pparg* in adult homeostatic skin ablates 99% of the SGs (*Veniaminova et al., 2023*). Interestingly, they showed that non-recombined cells from other parts of the hair follicle migrate to the SG zone and regenerate the genetically ablated SGs. This regeneration process is dependent on the hair growth cycle, with the SGs primarily regenerating during the anagen (active growth) phase and not in the telogen phase. While our studies cannot rule out the contribution of non-SG cells to the SG recovery seen upon the return of Notch activity, this recovery is not dependent on the hair growth cycle. Sebocytes are able to differentiate while the hair follicle is still in the telogen phase, indicating a lift of the

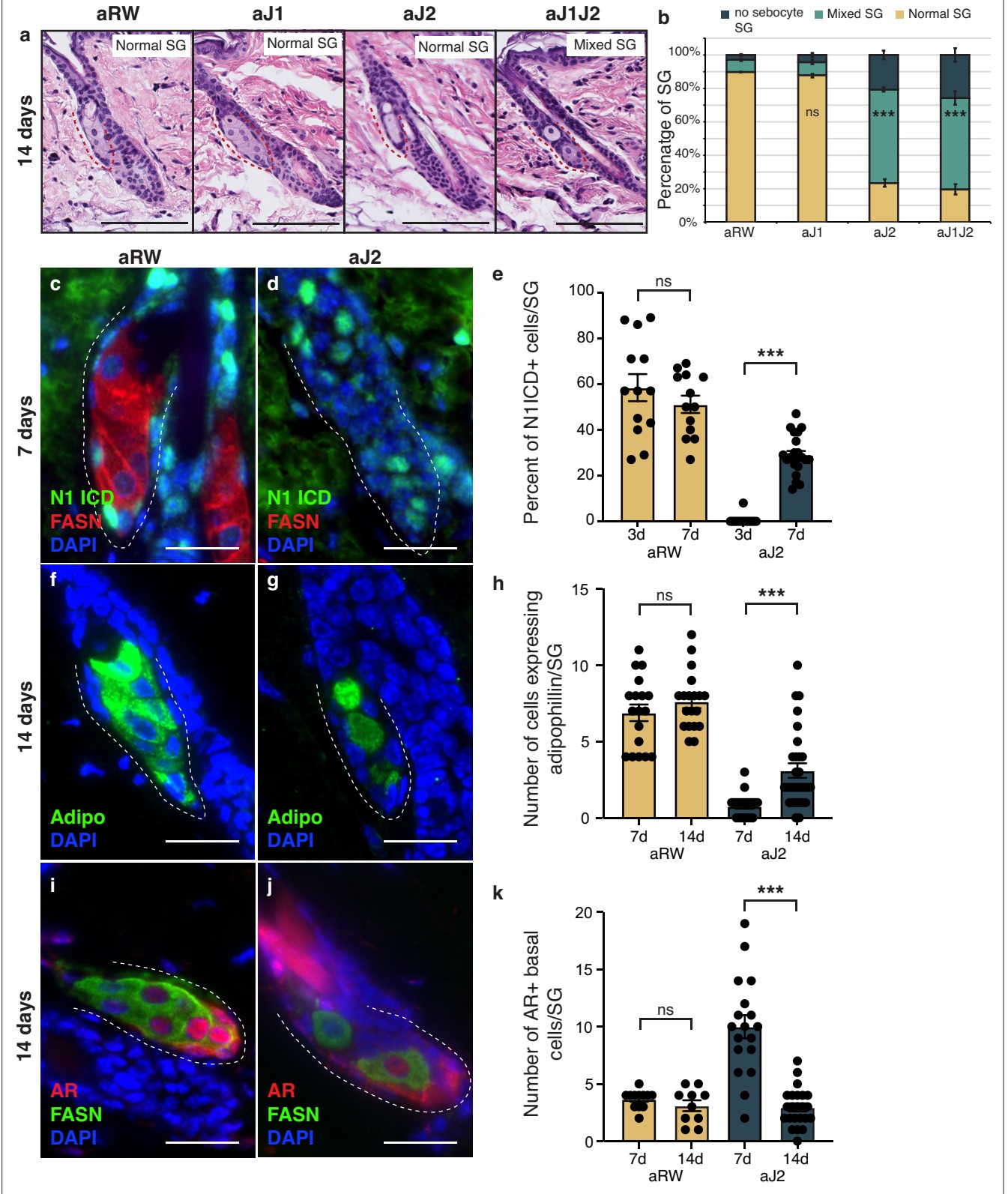

**Figure 5.** The block in sebocyte differentiation is lifted upon recovery of Notch activity. (**a**) Representative hematoxylin and eosin (H&E) images of sebaceous glands (SGs) from mice (n=5 each) treated with aRW, aJ1, aJ2, and aJ1J2, 14 d post treatment. (**b**) Quantification of type of SG found after each treatment. The SGs were divided into three categories: normal SGs containing a characteristic number of sebocytes (normal SG), SGs containing a mix of sebocytes and basal-like cells (mixed SG), and SGs containing no sebocytes and only basal-like cells (no sebocyte SG). p-values: aJ1=0.535,

*Figure 5 continued on next page*

*Figure 5 continued*

aJ2=2.90E-87, aJ1J2=1.73E-92. Chi-square test used for statistical analysis. All treatments were compared against aRW. Total n of SGs quantified per treatment: aRW = 485, aJ1=430, aJ2=394, aJ1J2=364. Scale bars are 100 µm. (**c,d**) Representative co-stain images for N1ICD and fatty acid synthase (FASN) in SGs from mice (n=5 each) treated with aRW (**c**) and aJ2 (**d**), 7 d post-treatment. (**e**) Quantification of the percentage of N1ICD + cells in SGs from mice (n=5 each) treated with aRW and aJ2, 3 and 7 d post-treatment. Percentage was calculated by dividing the number of N1ICD+ cells by the total number of basal-like cells in each SG. p-values: for comparison between day 3 and day 7 for aRW treatment = 0.297, for comparison between day 3 and day 7 for aJ2 treatment = 1.00E-14. Total n of SGs quantified per treatment: aRW at 3 d = 13, aRW at 7 d = 13, aJ2 at 3 d = 15, aJ2 at 7 d=22. (**f,g**) Representative adipophilin staining in SGs from mice (n=5 each) treated with aRW (**f**) and aJ2 (**g**), 14 d post treatment. (**h**) Quantification of the number of cells expressing adipophiln in each SG, 7 and 14 d after treatment with aRW, and aJ2. p-values: for comparison between day 7 and day 14 for aRW treatment = 0.292, for comparison between day 7 and day 14 for aJ2 treatment = 1.38E-4. Total n of SGs quantified per treatment: aRW at 7 d = 18, aRW at 14 d = 19, aJ2 at 7 d = 22, aJ2 at 14 d=29. (**i,j**) Representative co-stain images for androgen receptor (AR) and FASN in SGs from mice (n=5 each) treated with aRW (**i**) and aJ2 (**j**), 14 d post-treatment. (**k**) Quantification of the number of cells expressing AR in each SG, 7 and 14 d after treatment with aRW and aJ2. p-values: for comparison between day 7 and day 14 for aRW treatment = 0.273, for comparison between day 7 and day 14 for aJ2 treatment=9.53E-09. Total n of SGs quantified per treatment: aRW at 7 d=12, aRW at 14 d=10, aJ2 at 7 d=18, aJ2 at 14 d=23. Student's t-test used for statistical analysis. Error bars represent SEM. Scale bars are 25 µm.

The online version of this article includes the following source data and figure supplement(s) for figure 5:

**Source data 1.** Source data for *Figure 5*.

**Figure supplement 1.** The block in sebocyte differentiation is lifted upon recovery of Notch activity.

**Figure supplement 1—source data 1.** Source data for *Figure 5—figure supplement 1*.

differentiation block on the stem/progenitor cells. This reversibility of phenotype is intriguing from a therapeutic perspective, as it suggests a translational potential in skin disorders involving sebocyte overactivity, such as acne.

An important open question is how Notch signaling regulates sebocyte differentiation. FASN, which can be used as a readout for sebocyte differentiation, is a downstream target of AR (*Schirra et al., 2005*). We observed a Lrig1+/AR+/FASN- population in the normal SGs, similar to the basal-like cells that fill the finger-like remnants seen after Notch inhibition. This expression pattern indicates that AR requires a co-activator to activate downstream gene expression in the SG. Previous studies have reported that Notch effectors such as HEY1, HEY2, and HEYL can act as co-repressors of AR in prostate cells (*Belandia et al., 2005*; *Kamińska et al., 2020*; *Lavery et al., 2011*). It is possible that certain Notch effectors can act as either co-activators or co-repressors of AR in a context-dependent manner. Further investigation will be needed to understand the exact molecular role Notch signaling plays in regulating sebocyte differentiation.

# Methods

**Key resources table**

| Reagent type (species) or resource | Designation | Source or reference | Identifiers | Additional information |
|---|---|---|---|---|
| Biological sample (*Mus musculus*) | Dorsal skin tissue | Charles River-Hollister | | Females |
| Antibody | Anti-Jag1 (Mouse monoclonal) | Genentech | | Inhibiting antibody (20 mg/kg) |
| Antibody | Anti-Jag2 (Mouse monoclonal) | Genentech | | Inhibiting antibody (20 mg/kg) |
| Antibody | Anti-Notch1 (Human monoclonal) | Genentech | | Inhibiting antibody (5 mg/kg) |
| Antibody | Anti-Notch2 (Mouse monoclonal) | Genentech | | Inhibiting antibody (10 mg/kg) |

*Continued on next page*

| Reagent type (species) or resource | Designation | Source or reference | Identifiers | Additional information |
|---|---|---|---|---|
| Antibody | Anti-Ragweed (Mouse monoclonal) | Genentech | | Isotype control antibody – concentrations match the maximum dose of inhibiting antibodies |
| Antibody | Anti-N1ICD (Rabbit monoclonal) | Cell Signaling | Cat#: 4147 | Triple stain (20 ug/ml) IF (1:500) |
| Antibody | Anti-FASN (Mouse monoclonal) | BD | Cat#: 610963 | IF (1:100) |
| Antibody | Anti-Adipophilin (Guinea pig polyclonal) | Fitzgerald | Cat#: 20R-AP002 | IF (1:500) |
| Antibody | Anti-Ki67 (Rabbit monoclonal) | Thermo Fisher Scientific | Cat#: SP6 RM-9106-SO | IF (1:100) |
| Antibody | Anti-Lrig1 (Goat polyclonal) | R&D Systems | Cat#: AF3688-SP | IF (1:200) |
| Antibody | Anti-AR (Rabbit monoclonal) | Abcam | Cat#: ab133273 [EPR1535(2)] | IF (1:250) |
| Antibody | PowerVision poly-HRP anti-rabbit (Goat polyclonal) | Leica | Cat#: PV6119 | ready to use |
| Sequence-based reagent | RNAScope LS 2.5 Murine-*Jag2_C1* | ACD | Cat#: 417518 | nucleotides spanning from nt 552–1480 of reference sequence NM_010588.2 |
| Sequence-based reagent | RNAScope LS 2.5 Murine-*Notch1_C2* | ACD | Cat#: 404648-C2 | nucleotides spanning from nt 1153–1960 of reference sequence NM_008714.3 |
| Sequence-based reagent | RNAScope LS 2.0 Murine-*PPIB* probe | ACD | Cat#: 313917 | nucleotides spanning from nt 98~856 of reference sequence NM_011149.2 |
| Sequence-based reagent | RNAScope LS 2.0 *DapB* probe | ACD | Cat#: 312038 | nucleotides spanning from nt 414~862 reference sequence EF191515 |
| Commercial assay or kit | RNAscope LS Multiplex Reagent Kit | ACD | Cat#: 322800 | |
| Commercial assay or kit | Bond Epitope Retrieval Solution 2 | Leica | Cat#: AR9640 | |
| Commercial assay or kit | Opal Polaris 7 Auto Detection Kit | Akoya | Cat#: NEL811001KT | |

## Animal strains and treatments

Animals were maintained in accordance with the Guide for the Care and Use of Laboratory Animals (National Research Council, 2011). Genentech is an Association for Assessment and Accreditation of Laboratory Animal Care-accredited facility and all animal activities in this research study were conducted under protocols approved by the Genentech Institutional Animal Care and Use Committee. Mice were housed in individually ventilated cages within animal rooms maintained on a 14:10 hr light-dark cycle. Animal rooms were temperature and humidity-controlled, between 68–79°F (20.0 to 26.1°C) and 30 to 70%, respectively, with 10–15 room air exchanges per hour. Mice were fed Laboratory Autoclavable Rodent Diet 5010 (Lab Diet) and provided reverse osmosis purified drinking water ad libitum. Female

C57BL/6 mice were obtained from Charles River-Hollister, and were used for all experiments. Mice were housed under specific-pathogen-free conditions, and were 8 wk old upon treatment. This time point was chosen to correspond to the resting (telogen) phase of the hair growth cycle, as SG size can vary by hair cycle stage. All mice were injected intraperitoneally with a single dose of blocking antibodies diluted in sterile saline at the following concentrations: anti-Jag1 at 20 mg/kg, anti-Jag2 at 20 mg/kg, anti-Jag1 + anti-Jag2 at 20 mg/kg + 20 mg/kg=40 mg/kg, anti-Notch1 at 5 mg/kg, anti-Notch2 at 10 mg/kg and anti-Notch1 + anti-Notch2 at 5 mg/kg + 10 mg/kg=15 mg/kg. Anti-Ragweed isotype control antibody was injected at concentrations to match the maximum dose of treatment antibodies.

### Histopathological analysis and immunochemistry

Telogen mouse dorsal skin was collected at 3, 7, and 14 d post antibody injection. For frozen sections, skin samples were fixed in 4% paraformaldehyde in PBS, for 40 min at 4 °C, washed, and then immersed in 30% sucrose in PBS overnight at 4 °C. The tissue was then embedded in OCT (TissueTek), frozen immediately on dry ice, and stored at –80 °C. Additional skin tissue from the same animals was also fixed with 10% neutral buffered formalin and then used to create paraffin-embedded sections. Hematoxylin and eosin (H&E) staining was performed by the Pathology core at Genentech.

The triple immunofluorescence stain for Notch1 ICD (IHC), and Notch1 mRNA (ISH), and Jag2 mRNA (ISH) was performed by the histopathology development group at Genentech. ACD LS 2.5 probes were ordered from Advanced Cell Diagnostics. RNAScope LS 2.5 Murine-*Jag2_C1* (417518) nucleotides spanning from nt 552–1480 of reference sequence NM_010588.2, and Murine-*Notch1_C2* (404648-C2) nucleotides spanning from nt 1153–1960 of reference sequence NM_008714.3. For positive control, RNAScope LS 2.0 Murine-*PPIB* probe (313917) nucleotides spanning from nt 98~856 of reference sequence NM_011149.2 were used. For negative control RNAScope LS 2.0 *DapB* probe (312038) nucleotides spanning from nt 414~862 reference sequence EF191515 were used. For Immunohistochemistry, we used N1ICD (Cell Signaling, 4147), at 20 ug/ml working concentration.

The triple immunofluorescence for murine Jag2_Notch1 (dual ISH) with anti_Notch1 ICD (IHC) 3 Plex ISH_ISH_IHC in murine tissues using formalin-fixed, paraffin-embedded sections was performed on the Leica Bond-RX complete automation system using the RNAscope LS Multiplex Reagent Kit (322800). Slides were baked and dewaxed on Leica Bond-RX and pretreated with Bond Epitope Retrieval Solution 2 (ER2) (AR9640) from Leica at 100 °C for 40 min. After pretreatment, the probes were cocktailed, and hybridization was performed at 42 °C for 120 min followed by amplification steps and developed with Opal 570 fluor at 1:1000 and Opal 690 fluor at 1:1500 from Akoya using the Opal Polaris 7 Auto Detection Kit (NEL811001KT). The immunohistochemistry was performed upon completion of ISH. The primary antibody incubation was 60 min at room temperature, followed by secondary antibody incubation for 30 min at room temperature with HRP conjugated Goat anti-Rabbit (Perkin Elmer, NEF812001EA). The slides were then developed with Opal 780 fluor at 1:25 (Akoya, SKU FP1501001KT) following the manufacturer's instructions. Slides were imaged using Olympus VS200.

All other immunohistochemistry was performed using the following antibodies: N1ICD at 1:500 (Cell Signaling, 4147), FASN at 1:100 (BD, 610963), Adipophilin at 1:500 (Fitzgerald, 20R-AP002), Ki67 at 1:100 (Thermo Fisher Scientific, SP6 RM-9106-SO), Lrig1 at 1:200 (R&D Systems, AF3688-SP), and AR at 1:250 (Abcam, ab133273 [EPR1535(2)]). Slides were imaged using the Leica Thunder microscope.

Imaging parameters were identical for all images. Images were processed using Fiji and Adobe Illustrator.

## Acknowledgements

We are grateful to C Cottonham, S Hankeova, and G Hernandez for their helpful discussions. We thank the Genentech Research Pathology, Necropsy, and Histology laboratories for their experimental contributions. We appreciate the insightful feedback and comments on the paper from L Mosteiro, E Reyes, and B Biehs.

## Additional information

### Competing interests
Syeda Nayab Fatima Abidi: S.N.F.A. is an employee of Genentech. Sara Chan: S.C. is an employee of Genentech and holds shares in Roche. Kerstin Seidel: K.S. is an employee of Genentech and holds shares in Roche. Daniel Lafkas: D.L. was a Genentech employee, and is currently employed at Roche, and holds shares in Roche. Louis Vermeulen: L.V. is an employee of Genentech and holds shares in Roche. Frank Peale: F.P. is an employee of Genentech and holds shares in Roche. Christian W Siebel: C.W.S. was a Genentech employee, and holds shares in Roche. C.W.S is currently employed at Gilead Sciences.

### Funding
No external funding was received for this work.

### Author contributions
Syeda Nayab Fatima Abidi, Conceptualization, Data curation, Formal analysis, Validation, Investigation, Visualization, Writing – original draft; Sara Chan, S.C. developed the triple stain methodology, and performed the relevant staining; Kerstin Seidel, K.S. helped with some experimental design; Daniel Lafkas, The first observation of the phenotype came out of D.L.'s work; Louis Vermeulen, L.V. analyzed the data and contributed to writing the manuscript; Frank Peale, F.P. provided histopathological analysis and discussion; Christian W Siebel, C.W.S. directed and supervised the research, and analyzed the data

### Author ORCIDs
Syeda Nayab Fatima Abidi https://orcid.org/0000-0002-3390-9363

### Ethics
Animals were maintained in accordance with the Guide for the Care and Use of Laboratory Animals (National Research Council, 2011). Genentech is an Association for Assessment and Accreditation of Laboratory Animal Care-accredited facility and all animal activities in this research study were conducted under protocols approved by the Genentech Institutional Animal Care and Use Committee. Mice were housed in individually ventilated cages within animal rooms maintained on a 14:10-h light-dark cycle. Animal rooms were temperature and humidity-controlled, between 68 to 79 °F (20.0 to 26.1 °C) and 30 to 70%, respectively, with 10 to 15 room air exchanges per hour. Mice were fed Laboratory Autoclavable Rodent Diet 5010 (Lab Diet) and provided reverse osmosis purified drinking water ad libitum. Mice were housed under specific-pathogen-free conditions.

Reviewer #1 (Public review): https://doi.org/10.7554/eLife.98747.3.sa1
Reviewer #2 (Public review): https://doi.org/10.7554/eLife.98747.3.sa2
Reviewer #3 (Public review): https://doi.org/10.7554/eLife.98747.3.sa3
Author response https://doi.org/10.7554/eLife.98747.3.sa4

## Additional files

### Supplementary files
• MDAR checklist

### Data availability
All data generated or analysed during this study are included in the manuscript and supporting files; source data files have been provided for *Figure 1*; *Figure 2*; *Figures 3–5* and *Figure 1—figure supplement 1*, *Figure 2—figure supplement 1* and *Figure 5—figure supplement 1*.

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
