## [Editor Report · eLife assessment]

This work aimed at deconstructing how sebaceous gland differentiation is controlled in adult skin. Using monoclonal antibodies designed to inhibit specific Notch ligands or receptors, the authors present **convincing** evidence that the Jag2/Notch1 signaling axis is a crucial regulator of sebocyte progenitor proliferation and sebocyte differentiation. The **valuable** findings presented here contribute to the growing evidence that Notch signaling is not only key during the development of the skin and its appendages but also regulates cell fate in adult homeostatic tissues. From a translational perspective, it is intriguing that the effect of Jag2 or Notch1 inhibition, which leads to the accumulation of proliferative stem/progenitor cells in the sebaceous gland and prevents sebocyte differentiation, is reversible.

---

## [Referee Report · Reviewer #1 (Public review)]

Summary:

In this study, Abidi and colleagues used Notch pathway neutralizing antibodies to inhibit sebaceous glands in the skin. The authors find that blocking either the Notch1 receptor or the Jag2 ligand caused loss of the glands and increased retention of sebaceous progenitor cells. Moreover, these glands began to reappear 14 days after treatment.

Strengths:

Overall, this study definitively identifies the Notch receptor/ligand combination that maintains these glands in the adult. The manuscript is clearly written and the figures are beautifully made.

In this resubmitted manuscript, the authors have adequately addressed all the previous critiques.

---

## [Referee Report · Reviewer #2 (Public review)]

Summary:

In this report Abidi et al. use an antibody against Jag2, a Notch1 ligand, to inhibit its activity in skin. A single dose of this treatment leads to an impairment of sebocyte differentiation and an accumulation of basal sebocytes. Consistently Notch1 activity, measured as cleaved form of the Notch1 intracellular domain, is detected in basal sebocytes together with the expression of Jag2. Interestingly the phenotype caused by the antibody treatment is reversible.

Strengths:

The quality of the histological data with a clear phenotype, together with the quantification represents a solid base for the authors claims.

This work identifies that the ligand Jag2 is the Notch1 ligand required for sebocyte differentiation.

From a therapeutic point of view, it is interesting that the treatment with the anti-Jag2 is reversible.

Weaknesses:

The authors use a single approach to support their claims.

Future in vitro studies will be needed to understand how Notch signaling induces sebocyte differentiation (i.e. a cell-autonomous mechanism, a mechanism based on cell competition, etc.).

---

## [Referee Report · Reviewer #3 (Public review)]

Abidi et al. investigated the role of Notch signalling for sebaceous gland differentiation and sebocyte progenitor proliferation in adult mouse skin. By injecting antagonising antibodies against different Notch receptors and ligands into mice, the authors identified that the Notch1 receptor and, to a lesser extent, Notch2 receptor, as well as the Notch ligand Jagged2, contribute to the regulation of sebaceous gland differentiation. In situ hybridisation confirmed that treatment with anti-Jagged2 dramatically reduced the number of basal sebocytes staining for the transcriptionally active intracellular domain of Notch1. Loss of Notch activity in sebocyte progenitors robustly inhibited sebaceous gland differentiation. Under these conditions, the number of sebocyte progenitors marked by Lrig1 was not affected, while the number of proliferating basal sebocytes was increased. Upon recovery of Notch activity, sebaceous gland differentiation could likewise be recovered. By suggesting that Notch activity in sebocyte progenitors is required to balance proliferation and differentiation, these data bring valuable new and relevant findings for the skin field on the sebaceous gland homeostasis.

---

## [Author Response]

The following is the authors’ response to the original reviews.

We thank the editors and reviewers for their thoughtful comments on our manuscript. We greatly appreciated the suggestions and recommendations that helped us to improve the study. With adaptations, and inclusion of novel data and analyses, we have addressed all points raised, and hope that by these improvements the study further meets the standards for eLife.

**Reviewer #1 (Recommendations For The Authors):**
Minor text edits should be made.(1.1) As a recent study from the Wong lab also showed sebaceous gland regeneration following complete ablation (Veniaminova et al., 2023), this finding should be mentioned in the text, and the abstract ("Most strikingly...") should be toned down.

We thank the reviewer for the positive feedback, and for highlighting this part of the study from the Wong lab. Although we cited this study study in a different context, we had not discussed the sebaceous gland regeneration finding. We have now added this to the discussion section of the manuscript.

(1.2) Introduction: In lines 31-33 discussing the connection of sebaceous glands with skin disorders, the 5 references cited seem to replicate the citations from a similar sentence in Veniaminova et al., 2019. The authors should vary their citations, as there are likely other publications that can be cited here.

Additional references have been added.

**Reviewer #2 (Recommendations For The Authors):**
The manuscript is well written and the data are well presented in the figures.

We thank the reviewer for the positive feedback.

(2.1) Here are some points that could be taken into consideration to improve the manuscript:- Row 75 "the primary" regulator could be changed to "a crucial".

We appreciate this suggestion and have made the text edit.

- Row 86 could be added: ...is the dominant ligand of the Notch signalling.

We have made the text edit as suggested.

(2.2) Row 107-109 from the quantification of Figure 1G and Figure 2 it seems that only the aJ2 treatment has an SG phenotype. Why aJ1 doesn't have any effect? (same is true in other figures). If the data on aJ1 are maintained in the manuscript, this should be argued in the discussion section.

The reviewer is correct in noting that the aJ1 treatment does not cause the phenotype, and this is indeed one of the key findings of the study. This is maintained throughout the manuscript. We have also cited references showing that embryonic and adult deletions of *Jag1* do not cause any sebaceous gland defects. All these data argue that Jag1 is not the relevant Notch signaling ligand in sebocyte differentiation. We have further clarified this in the manuscript.

(2.3) Related to Figure 3G. As the Lrig1 stem cells can go towards both the sebocyte differentiation, or the sebaceous duct differentiation, it would be interesting to evaluate if the differentiation impairment caused by the antibody treatment affects in a similar manner (or not) the sebaceous duct differentiation. This could be tested through immunofluorescence, selecting markers of sebaceous duct.

We thank the reviewer for this thoughtful question. We are unable to find any unique markers of the sebaceous ducts (that are not expressed in other parts of the sebaceous gland, especially sebocytes) in the literature, thus, any analysis of markers would be confounded by its change of expression due to the loss of sebocytes.

However, we have evaluated the histology using bursting sebocytes releasing sebum as a proxy of a functional sebaceous duct. We have not found any significant differences between treatments using this metric (Fig. S1).

(2.4) As the word "therapeutic" is often underlined in the manuscript, maybe a few sentences on the transnational aspects of the results could be added to the discussion.

We thank the reviewer for highlighting this point. We have added this to the discussion.

(2.5) Figure 3 suggests that Jag2 is produced by basal sebocytes and used by these cells to induce sebocyte differentiation. I'm wondering if in an in vitro cell system (with a mixture of marked Jag2-expressing cells and marked Jag2-negative cells), it would be possible to understand if this mechanism of differentiation is a cell-autonomous mechanism or a mechanism based on cell competition (for instance, it would be possible that the progenitors compete for their niche on the basal layer by pushing neighbouring basal cells to differentiate presenting them Jag2).

We thank the reviewer for the insightful suggestion. The mechanistic underpinning of how Notch signaling induces sebocyte differentiation is still unclear, and we find the reviewer’s suggestion very interesting. However, establishing an *in vitro* model that captures the aspects mentioned, would require a lot of optimization and validation. To help rapid dissemination of our findings we elected to keep this out of the manuscript, but we will certainly consider it for future studies.

**Reviewer #3 (Recommendations For The Authors):**
(3.1) The authors focussed on mouse back skin sebaceous glands to analyse the phenotype. Are the effects also reproducible in the sebaceous glands of the mouse ears and tail epidermis? If so, the data should be strengthened by quantifying the phenotype using tail epidermal whole mounts (Braun et al., 2003; Development, PMID: 12954714), ideally by co-staining sebaceous glands for differentiation markers (e.g. FASN, Adipophilin) or lipid deposits (e.g., Oil red O). Also, the authors need to clarify how many sebaceous glands were scored per mouse. If not, please provide a rationale explaining the location restriction.

We thank the reviewer for pointing this out. Indeed, we have only incorporated data from the telogen dorsal skin of the animals. We have now more accurately reflected this in the revised manuscript. Additionally, we have added the number of sebaceous glands quantified in each figure per the reviewer’s suggestion.

Since the stage of hair growth cycle can affect the sebaceous glands, we chose the resting (telogen) phase of the hair cycle to reliably study the sebaceous glands. At 8 weeks of age, hair follicles have uniformly entered the telogen phase. As subsequent re-entry into the anagen phase is asynchronous in the adult skin, the color of the dorsal skin of C57BL/6 mice can be used to determine whether the hair follicles are in the telogen phase or not. These reasons led us to choose this location, allowing us to study only telogen phase hair follicles.

We also point out that previously reported data (Estrach et al., 2006) did not show differences between dorsal and tail skin, so we assume the mechanisms must largely be conserved. However, as the reviewer rightfully points out, we cannot be sure and have, therefore, indicated the dorsal location throughout the manuscript.

(3.2) The micrographs in Figure 2 suggest that expression of both Jagged2 and Notch1 (intercellular domain) is not restricted to the sebaceous glands, as both molecules appear to be detected also in the isthmus and lower hair follicle. Of note, the online tool provided by the Kasper and Linnarsson labs (http://linnarssonlab.org/epidermis/) shows that both molecules are more widely expressed in mouse back skin. Please provide some analysis of the overall expression of these molecules in mouse skin. In line, is the observed effect of using the antagonising antibodies restricted to the sebaceous glands? Please provide additional data on proliferation and differentiation in the interfollicular epidermis, hair follicle cycling, and other skin compartments. For instance, the data published in the cited paper by Lafkas et al. (2005) suggest a thickening of the dermal adipocyte layer upon Jagged2 inhibition using monoclonal therapeutic antibodies.

The reviewer is correct in noting that expression of both Jag2 and Notch1 is not restricted to the sebaceous gland. The Notch signaling pathway is a well-known regulator for epidermal differentiation, and members of the pathway are expressed in various locations of the skin, including the interfollicular epidermis and the hair follicle. The expression and function of Notch signaling in these locations has been reviewed in (Hsu et al., 2014; Nowell and Radtke, 2013; Watt et al., 2008). We have also added zoomed out images showing expression of Jag2 and Notch1 in the skin (Figure S2e,f).

The effect of the antagonizing antibodies is not restricted to sebaceous glands, as we already noted in our discussion section: “While injections of the Notch blocking antibodies are systemic, we only observed a reduction in the number of Notch-active cells in the IFE, but not a complete loss.” The functional impact of the antibodies is likely beyond the sebaceous gland, as the reviewer points out, but understanding the full effect in other compartments, we consider beyond the scope of the current study.

In our previous study (Lafkas et al., 2015), the skin was examined at different animal ages/gender and using different antibody dosing regimens, which is the likely explanation for the differences observed. We have now quantified the width of the adipocyte layer and the IFE and show that there are no significant differences between treatments (Figure S1g-j). This together with the histology suggest that there are no significant differences in the differentiation and proliferation of these compartments.

(3.3) Since Jagged1 is a Wnt/beta-catenin target gene that is essential for (ectopic) hair follicle formation and differentiation (Estrach et al., 2006, Development, PMID: 17035290) and the sebaceous gland is widely considered as an epidermal compartment with absent/low Wnt/beta-catenin pathway activity during normal homeostasis (Lim & Nusse, 2013, Cold Spring Habor Perspectives in Biology, PMID: 23209129), how is the expression of Notch1 and Jagged2 regulated upstream in sebocyte progenitors? It would be important to bring some more mechanistic insights into the upstream regulation of Notch activity. In line with comment 2, how are the compartment-specific effects molecularly regulated if the effects are not restricted to the sebaceous glands?

The reviewer is correct in noting that the Wnt pathway does not seem to be a likely candidate for driving sebocyte differentiation through Notch signaling. Indeed, Wnt inhibition is required for sebocyte differentiation (Merrill et al., 2001; Niemann et al., 2002), and the *Jag2* promoter region also does not contain TCF binding sites (Katoh and Katoh, 2006).

We speculate that Myc might regulate Notch signaling in the sebaceous gland. It is expressed in the sebaceous gland basal stem cells and has been reported to positively regulate sebocyte differentiation (Cottle et al., 2013). In addition, studies have shown that *Jag2* is a Myc target gene (Fiaschetti et al., 2014; Yustein et al., 2010). However, evaluating which upstream pathway potentially regulates Notch signaling, and resolving the regulatory network of sebocyte differentiation beyond the direct Notch ligands and receptors would require extensive *in vivo* modeling using KO and transgenic animals, which we consider to be beyond the scope of the current manuscript.

References

Cottle DL, Kretzschmar K, Schweiger PJ, Quist SR, Gollnick HP, Natsuga K, Aoyagi S, Watt FM. 2013. c-MYC-Induced Sebaceous Gland Differentiation Is Controlled by an Androgen Receptor/p53 Axis. *Cell Rep* 3:427–441. doi:10.1016/j.celrep.2013.01.013

Estrach S, Ambler CA, Celso CLL, Hozumi K, Watt FM. 2006. Jagged 1 is a β-catenin target gene required for ectopic hair follicle formation in adult epidermis. *Development* 133:4427–4438. doi:10.1242/dev.02644

Fiaschetti G, Schroeder C, Castelletti D, Arcaro A, Westermann F, Baumgartner M, Shalaby T, Grotzer MA. 2014. NOTCH ligands JAG1 and JAG2 as critical pro-survival factors in childhood medulloblastoma. *Acta Neuropathol Commun* 2:39. doi:10.1186/2051-5960-2-39

Hsu Y-C, Li L, Fuchs E. 2014. Emerging interactions between skin stem cells and their niches. *Nat Med* 20:847–856. doi:10.1038/nm.3643

Katoh Masuko, Katoh Masaru. 2006. Notch ligand, JAG1, is evolutionarily conserved target of canonical WNT signaling pathway in progenitor cells. *Int J Mol Med*. doi:10.3892/ijmm.17.4.681

Lafkas D, Shelton A, Chiu C, Boenig G de L, Chen Y, Stawicki SS, Siltanen C, Reichelt M, Zhou M, Wu X, Eastham-Anderson J, Moore H, Roose-Girma M, Chinn Y, Hang JQ, Warming S, Egen J, Lee WP, Austin C, Wu Y, Payandeh J, Lowe JB, Siebel CW. 2015. Therapeutic antibodies reveal Notch control of transdifferentiation in the adult lung. *Nature* 528:127–131. doi:10.1038/nature15715

Merrill BJ, Gat U, DasGupta R, Fuchs E. 2001. Tcf3 and Lef1 regulate lineage differentiation of multipotent stem cells in skin. *Genes Dev* 15:1688–1705. doi:10.1101/gad.891401

Niemann C, Owens DM, Hülsken J, Birchmeier W, Watt FM. 2002. Expression of ΔNLef1 in mouse epidermis results in differentiation of hair follicles into squamous epidermal cysts and formation of skin tumours. *Development* 129:95–109. doi:10.1242/dev.129.1.95

Nowell C, Radtke F. 2013. Cutaneous Notch Signaling in Health and Disease. *Cold Spring Harb Perspect Med* 3:a017772. doi:10.1101/cshperspect.a017772

Watt FM, Estrach S, Ambler CA. 2008. Epidermal Notch signalling: differentiation, cancer and adhesion. *Curr Opin Cell Biol* 20:171–179. doi:10.1016/j.ceb.2008.01.010

Yustein JT, Liu Y-C, Gao P, Jie C, Le A, Vuica-Ross M, Chng WJ, Eberhart CG, Bergsagel PL, Dang CV. 2010. Induction of ectopic Myc target gene JAG2 augments hypoxic growth and tumorigenesis in a human B-cell model. *Proc Natl Acad Sci* 107:3534–3539. doi:10.1073/pnas.0901230107